# Fast and accurate text classification: skimming, rereading and early stopping

**Keyi Yu**
Department of Computer Science
University of Illinois at Urbana-Champaign
Tsinghua University
yu-ky14@mails.tsinghua.edu.cn

**Yang Liu**
Department of Computer Science
University of Illinois at Urbana-Champaign
liu301@illinois.edu

**Alexander G. Schwing**
Department of Electrical and Computer Engineering
University of Illinois at Urbana-Champaign
aschwing@illinois.edu

**Jian Peng**
Department of Computer Science
University of Illinois at Urbana-Champaign
jianpeng@illinois.edu

## Abstract

Recent advances in recurrent neural nets (RNNs) have shown much promise in many applications in natural language processing. For most of these tasks, such as sentiment analysis of customer reviews, a recurrent neural net model parses the entire review before forming a decision. We argue that reading the entire input is not always necessary in practice, since a lot of reviews are often easy to classify, i.e., a decision can be formed after reading some crucial sentences or words in the provided text. In this paper, we present an approach of fast reading for text classification. Inspired by several well-known human reading techniques, our approach implements an intelligent recurrent agent which evaluates the importance of the current snippet in order to decide whether to make a prediction, or to skip some texts, or to re-read part of the sentence. Our agent uses an RNN module to encode information from the past and the current tokens, and applies a policy module to form decisions. With an end-to-end training algorithm based on policy gradient, we train and test our agent on several text classification datasets and achieve both higher efficiency and better accuracy compared to previous approaches.

## 1 Introduction

Recurrent neural nets (RNNs), including GRU nets (Chung et al., 2014) and LSTM nets (Hochreiter & Schmidhuber, 1997), have been increasingly applied to many problems in natural language processing. Most of the problems can be divided into two categories: sequence to sequence (seq2seq) tasks (Sutskever et al., 2014) (e.g., language modeling (Bengio et al., 2003; Mikolov et al., 2010), machine translation (Cho et al., 2014; Bahdanau et al., 2014; Kalchbrenner & Blunsom, 2013), conversational/dialogue modeling (Serban et al., 2016), question answering (Hermann et al., 2015; Weston et al., 2015; Lee et al., 2016), and document summarization (Rush et al., 2015; Nallapati et al., 2016)); and the classification tasks (e.g., part-of-speech tagging (Santos & Zadrozny, 2014), chunking, named entity recognition (Collobert et al., 2011), sentimental analysis (Socher et al., 2011), and document classification (Kim, 2014; Sebastiani, 2002)). To solve these problems, models often need to read every token or word of the text from beginning to the end, which is necessary for most seq2seq problems. However, for classification problems, we do not have to treat each individual word equally, since certain words or chunks are more relevant to the classification task at hand. For instance, for sentiment analysis it is sufficient to read the first half of a review like "this movie is amazing" or "it is the best I have ever seen," to provide an answer even without reading the rest of the review. In other cases, we may want to skip or skim some text without carefully checking it. For example, sentences such as "it's worth to try" are usually more important than irrelevant text such as "we got here while it's still raining outside" or "I visited on Saturday." On the other hand, sometimes, we want to re-read some sentences to figure out the actual hidden message of the text.

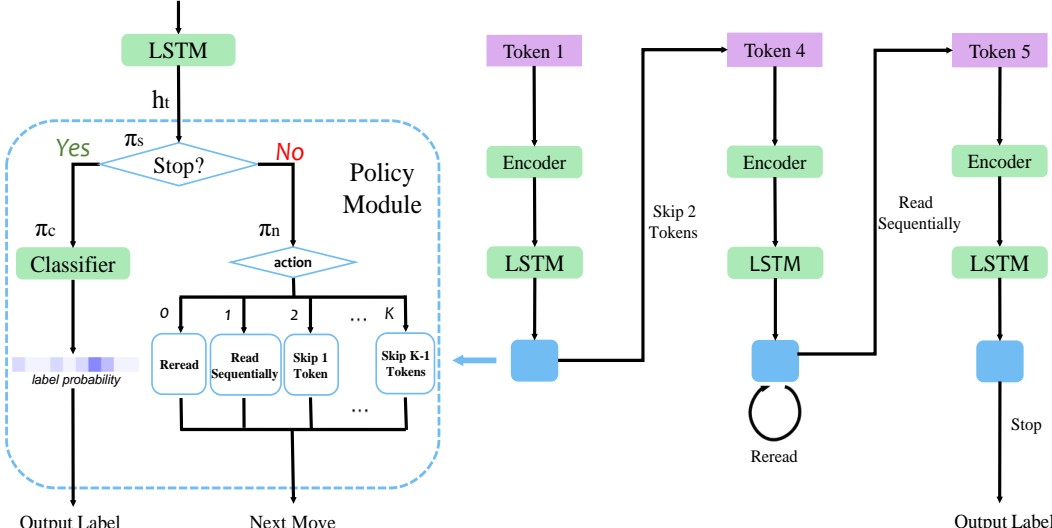

Figure 1: Outline of the proposed model.

All of these techniques enable us to achieve fast and accurate reading. Similarly, we expect RNN models to intelligently determine the importance or the relevance of the current sentence in order to decide whether to make a prediction, whether to skip some texts, or whether to re-read the current sentence.

In this paper, we aim to augment existing RNN models by introducing efficient partial reading for classification, while maintaining a higher or comparable accuracy compared to reading the full text. To do so, we introduce a recurrent agent which uses an RNN module to encode information from the past and the current tokens, and applies a policy module to decide what token to read next (e.g., rereading the current token, reading the next one, or skipping the next few tokens) or whether the model should stop reading and form a decision. To encourage fast and accurate reading, we incorporate both classification accuracy and the computational cost as a reward function to score classification or other actions made by the agent during reading. We expect that our agent will be able to achieve fast reading for classification with both high computational efficiency and good classification performance. To train this model, we develop an end-to-end approach based on the policy gradient method which backpropagates the reward signal into both the policy module (also including the classification policy) and the recurrent encoder.

We evaluate our approach on four different sentiment analysis and document topic classification datasets. By comparing to the standard RNN models and a recent LSTM-skip model which implements a skip action (Yu et al., 2017), we find that our approach achieves both higher efficiency and better accuracy.

## 2  METHODS

### 2.1  MODEL OVERVIEW

Given an input sequence $x_{1:T}$ with length $T$, our model aims to predict a single label $y$ for the entire sequence, such as the topic or the sentiment of a document. We develop a technique for skimming, re-reading, early stopping and prediction, with the goal of (i) skipping irrelevant information and reinforcing the important parts, and (ii) to enable fast and accurate text classification. Specifically, the model will read the current token/chunk $x_{i_t}$ at time step $t$, encode the data $x_{i_t}$ and previous information $h_{t-1}$ into a feature $h_t$, and then decide the next token to read by skimming/skipping or to stop to form a final prediction (see Figure 1). Such a model can be fully defined on top of a RNN structure and trained in an end-to-end fashion via back-propagation of a well defined reward signal. Both skimming and re-reading actions can be defined similarly by first choosing a step size

$k \in \{0, 1, 2, \cdots, K\}$ and then setting $i_{t+1} = i_t + k$. When $k = 0$, the model rereads the current token; when $k = 1$, the model moves to the next token sequentially; when $k > 1$, the model skips the next $k - 1$ tokens. If the current action is to stop or the next token to read is after the last token of the input sequence text, the model will stop and output a label. All of these actions are defined by a policy module $\Pi$ which takes the recurrent feature $h_t$ as input and outputs a stop signal and a label or generates a step size $k$ and moves to the next token $x_{i_{t+1}=i_t+k}$.

## 2.2   MODEL SPECIFICATION AND TRAINING

The design of the policy module $\Pi$ plays an critical role in our framework. It should (i) read as much significant text as possible to ensure a confident classification output and (ii) be computationally efficient, e.g., avoiding reading to the end of the text if sufficient information is already obtained and skipping irrelevant or unimportant part of the text. More specifically, for each step, the policy module $\Pi$ should decide whether the information collected is convincing enough to stop reading and make a prediction. Otherwise it will need to evaluate the importance of the current semantic unit or token just read to decide which token to be read in the next step. By formulating this process as a sequential decision process, at each time step $t$, the policy module takes the output $h_t$ of an encoder, which summarizes the text read before and the current token $x_{i_t}$, and outputs a probability distribution $\pi_t$ defined over actions. It is worth noting that to save computation, the actions are determined only by the latent representation $h_t$. At each time step $t$, a sequence of actions are generated by first sampling a stopping decision in the form of a binary variable $s$ from a Bernoulli distribution $\pi_S(\cdot|h_t)$. If $s = 1$, the model stops and draws a label $\hat{y}$ from a conditional multinomial distribution specified by a classifier $\pi_C(\cdot|h_t, s = 1)$; otherwise, the model draws a step size $k \in \{0, \ldots, K\}$ from another conditional multinomial distribution $\pi_N(\cdot|h_t, s = 0)$ to jump to the token $x_{i_{t+1}=i_t+k}$.

Thus the probability of a sequence of actions that reads text $X_{i_1:i_t} = \{x_{i_1}, x_{i_2}, ..., x_{i_t}\}$, stops at time $t$, and outputs a label $\hat{y}$ can be written as the joint distribution

$$\Pi(X_{i_1:i_t}, \hat{y}) = \pi_S(s = 1|h_t)\pi_C(\hat{y}|h_t, s = 1)\prod_{j=1}^{t-1}\pi_S(s = 0|h_j)\pi_N(k_j = i_{j+1} - i_j|h_j, s = 0),$$

or simply as

$$\Pi(X_{i_1:i_t}, \hat{y}) = \pi_S(1|h_t)\pi_C(\hat{y}|h_t, 1)\prod_{j=1}^{t-1}\pi_S(0|h_j)\pi_N(k_j|h_j, 0). \tag{1}$$

Hereby, $k_j = i_{j+1} - i_j$ is the step size sampled at time $j$ which ensures the model moves from token $x_{i_j}$ to $x_{i_{j+1}}$, and $h_j = RNN(x_{i_j}, h_{j-1})$ is computed by the RNN module.

To encourage fast and accurate text reading, we want to minimize the difference between true label and predicted while ensuring a low computational cost, which is measured by the length of the assessed text. Hence, as the reward for the last output action, we use $-\mathcal{L}(\hat{y}, y)$, where $\mathcal{L}$ is a loss function that measures the accuracy between predicted label $\hat{y}$ and true label $y$. For other actions we use a negative computational cost $-\mathcal{F}$. Hereby, $\mathcal{F}$ is the normalized FLOP count used at each time step which is approximately constant. Note that the FLOP count for the last step, $F_t$, differs, since it also includes the cost of the classification. Overall, the reward signal is defined as:

$$r_j = \begin{cases} -\mathcal{L}(\hat{y}, y) - \alpha\mathcal{F}_t & \text{if } j = t \text{ is the final time step} \\ -\alpha\mathcal{F} & \text{otherwise} \end{cases},$$

where $\alpha$ is a trade-off parameter between accuracy and efficiency.

Assume that the entire policy $\Pi_\theta$ is parameterized by $\theta = \{\theta^{\pi_S}, \theta^{\pi_C}, \theta^N, \theta^{RNN}\}$, where $\theta^{RNN}$ subsumes the parameters for the encoder. Our final goal is to find the optimal $\theta$ which maximize the expected return defined by:

$$J(\theta) = \mathbb{E}_{(x,y)\sim\mathcal{D}}\left[\sum_t \mathbb{E}_{(X_{i_1:i_t}, \hat{y})\sim\Pi}\sum_{j=1}^t \gamma^{j-1}r_j\right], \tag{2}$$

where the first summation is used for integrating all possible sequences with different lengths to ensure the normalization of the distribution $\Pi$, and $\gamma \in (0, 1)$ is a discount factor. It is not hard to

see that $J$ is infeasible to compute by enumerating all possibilities in the summation and expectation. Fortunately, we can apply the policy gradient algorithm (Williams, 1992) to optimize this objective by estimating the gradient using Monte Carlo rollout samples, without doing expensive integration or enumeration. The REINFORCE policy gradient of the objective on data $(x, y)$ can be derived as follows:

$$\widehat{\nabla_\theta J} = \nabla_\theta [\log \pi_S(1|h_t) + \log \pi_C(\hat{y}|h_t, 1) + \sum_{j=1}^{t-1} (\log \pi_S(0|h_j) + \log \pi_N(k_j|h_j, 0))] \sum_{j=1}^{t} \gamma^{j-1} r_j.$$

Considering that the length of the rollout sequence can differ significantly, the space for policy exploration is very large, thus making the variance of the gradient estimation very high. To remedy this, we also implement the advantage actor-critic algorithm (Konda & Tsitsiklis, 2000), which couples partial future return with each action and estimates a value function as the baseline for variance reduction. We find this procedure to provide better performance than the vanilla REINFORCE algorithm.

It is worth noting that this policy gradient method eventually is able to backpropagate both classification accuracy and computational cost signals to every module in our model, including the stopping/skipping distributions, the label distribution and even the recurrent encoder, thus providing an end-to-end solution to text classification problems.

Overall, our model aims to accelerate text classification while still achieving a high accuracy. The hyperparameter $\alpha$ is used to control the trade-off between accuracy and time cost. If we set $\alpha$ to be a relatively large value, our model will be more boldly to skip tokens, stop reading and output a label. If $\alpha$ is small, our model would like to (re)read more tokens. Actually, the reward for penalizing the computational cost can be seen as a Lagrangian multiplier which is used to constrain the average cost of the computation. Therefore, there is a mapping between $\alpha$ and the amortized computational budget allocated for each sample. Given a budget, we can tune $\alpha$ to provide a model with best classification accuracy with the amortized cost within the budget. This is desirable for many cost-sensitive applications, such as those on mobile devices.

## 3 EXPERIMENTS

In this section, we illustrate our approach using two representative text classification tasks: sentiment analysis and topic classification. To perform a solid demonstration on re-reading and skimming, we conduct experiments on three different syntactic levels. We will first introduce the results on the word level before discussing character and sentence level performance.

**General Experimental Settings:** In our experiments, we use the IMDB and Yelp dataset for sentiment analysis, and the AG news and DBpedia for topic classification. To evaluate each classifier, we use predictive accuracy as the performance metric and average per-data floating point operations (FLOPs) as the computational cost metric. We also take the FLOPs of the policy module into account, even though they are much lower than the classifier. The energy cost for the policy module is about 1 to 2 million FLOPs per sentence, which is much smaller than the total FLOPs needed for the recurrent module and the classifier.

**Hyper-parameters:** We use the Adam (Kingma & Ba, 2014) optimizer with a learning rate of 0.001 in all experiments. For the recurrent network structure, we use a convolution layer with 128 kernels of size 5 and stack it as input to an LSTM with a hidden size of 128. For $\pi_S$ and $\pi_N$ policy network, we use a three hidden-layer MLP with 128 hidden units per layer. For $\pi_C$ and value network, we use a single-layer MLP with 128 hidden units. For all experiments, the maximal step size $K$ is set to 3.

### 3.1 RESULTS

We first evaluate our method on the IMDB movie dataset (Maas et al., 2011). We randomly split it into 20,000 training, 5,000 validation and 25,000 test samples. The average length in the dataset is 240 words. We adopt the same preprocessing method as Yu et al. (2017), either padding or

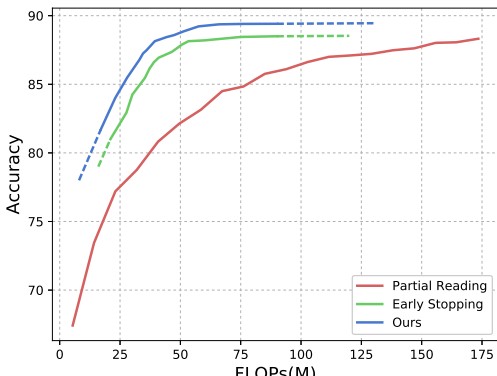

Figure 2: Comparison of partial reading, early stopping and our model on the IMDB dataset. Curves are obtained by changing the computational budget for each method. For Ours and Early Stopping, we adjust the parameter $\alpha$.

truncating each sentence to 400 words. We use a chunk-size of 20 words, i.e., at each step, the classifier reads 20 words. When the action is rereading or skipping, it rereads or skips several chunks of 20 words.

To demonstrate the effectiveness of re-reading and skimming, we design three baseline models: (1) The *early stopping model*, which has only a stopping module to decide when to terminate reading the paragraph, the classifier and policy module are jointly trained on the entire training corpus; (2) The *partial reading model*, which is a classifier with same architecture trained on the truncated sentences decided by the stopping model (same as the one in the early stopping model. Thus, although the *partial reading model* has the same computational budget as the *early stopping model*, the prediction performance may differ; (3) The *whole reading model*, which tries to use the whole corpus as training data.

Figure 2 shows our comparison on the IMDB dataset, where the blue line indicates our proposed model while green and red one denote *early-stopping model* and *partial reading model*, respectively. The x-axis denotes the FLOP count (in millions) and the y-axis indicates the accuracy. Here the FLOP count is determined by the choice of the hyper-parameter $\alpha$. As $\alpha$ increases, we obtain a curve indicating the trade-off between accuracy and energy cost. From this plot, we observe that both blue line and green line outperform the red line significantly. In addition, rereading and skipping further improve the performance of the model with only the early stopping mechanism. This observation implies that training the classifier jointly with the policy model improves both computational efficiency and accuracy.

Besides the word-level evaluation, we also conduct experiments on a smaller scale syntactic unit: character-level. In detail, we perform topic classification on two large-scale text datasets (Zhang et al., 2015): the AG news dataset contains four topics, 120,000 training news, 10,000 validation news, 7,600 testing news. The DBpedia dataset contains 14 topics, 560,000 training entities, 10,000 validation entities and 70,000 testing entities. The results are summarized in Figure 3. We observe that our proposed model outperforms the partial reading baseline by a significant margin.

Furthermore, we evaluate our proposed model on a larger syntactic level: sentence level. We use Yelp review sentiment analysis dataset for this experiment. The Yelp dataset includes 500,000 training reviews, 20,000 validation reviews and 40,000 testing reviews. To evaluate on the larger semantic unit, we treat each sentence as a *token*, which will be read sequentially by the RNN encoder. The performance is provided in Figure 4. We observe that our proposed model achieves superior performance while being significantly faster.

We summarize the obtained performance improvements in Table 1. On four different datasets and for three different syntactic levels we observe significant speedups when using the proposed techniques for skimming, rereading and early stopping, while maintaining the accuracy. A partial reading model

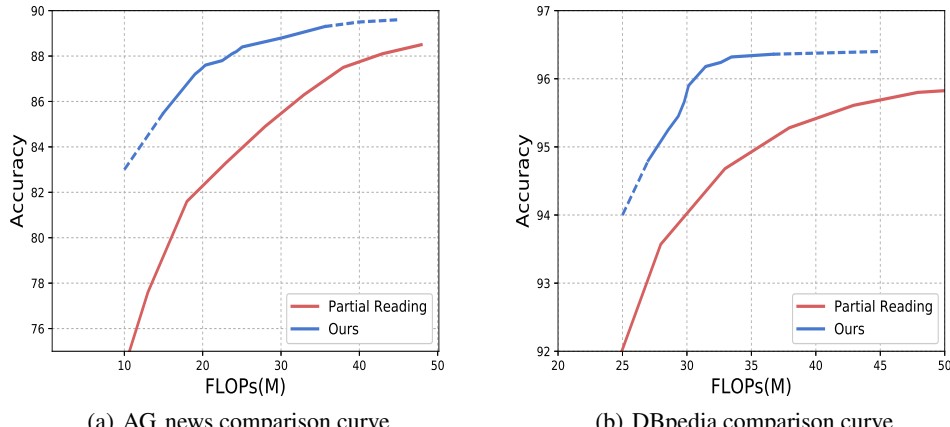

(a) AG_news comparison curve       (b) DBpedia comparison curve

Figure 3: Comparisons on character-level topic classification on two datasets: The x-axis and y-axis are representing FLOPs and accuracy, respectively. The red line denotes the partial reading baseline. The blue line indicates our proposed model.

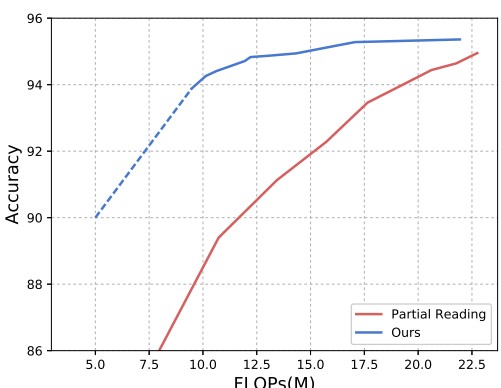

Figure 4: Comparison of the partial reading model and our model on the Yelp challenge dataset.

which has the same computational cost achieves results that are less accurate, which illustrates the benefits of a flexible model. In addition, our model achieves about 0.5-1 percent accuracy improvement compared to the full-reading model.

Finally, we compare our model to a recently published baseline (Yu et al., 2017), which only implements the skipping actions with $k \in \{1, 2, ..., K\}$ but without rereading, and simply do early stopping when $k = 0$. We implemented their algorithm for a fair comparison. Results in Table 2 show that our model is much more efficient than their LSTM-skip model at the same-level of accuracy, which is marginally better than full reading baseline. These results demonstrated that our proposed rereading and skimming mechanisms are effective on a variety of text classification tasks including sentiment analysis and topic classification. Also it is effective on different level of semantics: character-level, word-level or even sentence-level. With the help of our mechanisms, we could achieve both higher accuracy and faster speed.

## 3.2 ABLATION ANALYSIS

In this section, we conducted an ablation study to demonstrate the effectiveness of each action mechanism in our method: skimming, rereading and early-stopping. Our experiment was performed

| Syntactic level | Dataset | Speedup | Accuracy | Relative PR Accuracy |
|:---:|:---:|:---:|:---:|:---:|
| **Word** | IMDB | **4.11x** | 88.32% | -7.19% |
| **Character** | AG_news | **1.85x** | 88.50% | -4.42% |
| | DBpedia | **2.42x** | 95.99% | -1.94% |
| **Sentence** | Yelp | **1.58x** | 94.95% | -3.38% |

Table 1: Summary of our results: We compare our model to a complete reading baseline and a partial reading (PR) baseline on four datasets and three different syntactic levels. Here we report the speedups of our model compared to whole-reading baseline at the same accuracy level. Also we report the relative performance of the partial reading baseline with the same computational cost as our model.

| Syntactic-level | Dataset | Accuracy | FLOPs (Yu et al., 2017) | FLOPS (Ours) |
|:---:|:---:|:---:|:---:|:---:|
| **Word** | IMDB | 88.82% | 57.80% | **29.33%** |
| **Character** | AG_news | 88.60% | 87.68% | **57.55%** |
| | DBpedia | 96.21% | 76.35% | 44.34% |
| **Word** | Yelp | 95.14% | 82.34% | 70.02% |

Table 2: Summary of our results: We compare our model to (Yu et al., 2017), showing the relative FLOPs necessary to achieve the same accuracy on two datasets used in both theirs and this paper.

on the word-level IMDB dataset, and the result is presented in Figure 5. The blue curve denotes the performance of the model with all actions (skimming, rereading and early-stopping) enabled. The green one denotes the performance of the model with only the early-stopping actions. Between these two curves, the red curve represents a model with rereading and early-stopping action, and the yellow line represents a model with skimming and early-stopping actions. Note that the performance of the green curve is the worst, indicating that rereading and skimming mechanisms are necessary. Furthermore, the blue curve is better than all other ones, indicating that combining skimming and rereading together can further improve the performance of the policy model.

## 3.3 CASE STUDIES

To obtain a more detailed understanding of our model, we first show the actions taken by our model on a sentiment analysis example (Figure 6), on which the LSTM full-reading model failed to give the right classification. We show the degree of positiveness given by LSTM model encoded in color, from green representing positiveness to brown representing negativeness.

The paragraph starts with a sentence with strong positiveness of a dinner, then followed by a few sentences on some confusing description on the dinner. Many trivial or even negative words show up in the explanation. As a result, the output of the full-reading model gradually changes from positive to negative and finally results in a negative signal. Importantly, after our model reads the first two sentences, the policy module decides that it is confident enough to make a decision yielding the correct answer.

Next we illustrate how the rereading and skimming actions are useful to identify important information in the text. As shown in Figure 7, our model first reads a key word "stake" and is confident that the document is about money. Then it skims a few irrelevant tokens to read about "buying stake in Biotechnology" the following two tokens. The phrase "5 percent stake" showed up twice. Our model consider it to be important so it re-reads this token. At this time, the model basically knows this text is about business with a reasonable confidence. Then it skips to read about "collaboration deal" and stops to make a confident prediction.

## 4 RELATED WORK

The idea of improving time efficiency with adaptive computation has been studied extensively and throughout the years Weiss & Taskar (2013). For example, the adaptive computation time algorithm Graves (2016) on recurrent neural networks proposed to utilize early stopping action to save

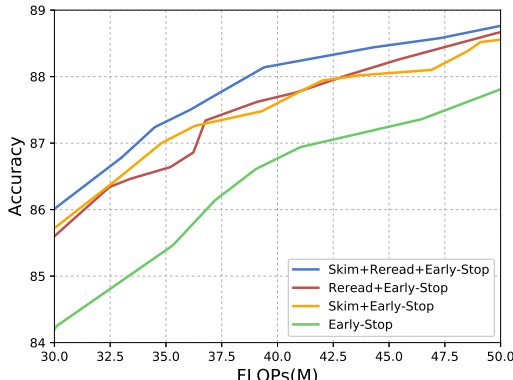

Figure 5: Comparison between different action combination settings to demonstrate the effectiveness of each mechanism: the blue line denotes our model with all actions, the green line denotes the model with only an early-stopping module. Between these two lines, the red one represents the model with rereading and early-stopping (no skimming) while the yellow one represents the model with skimming and early-stopping (no rereading).

STOP

What a great dinner! I had the pasta trio special. They walk around with three pastas and just re-fill your plate. I even wanted one of them without an ingredient , and they did not bat an eye. They even had the pasta prepared the same way to refill me! I haven't been back, and I kick myself for it on my way out of town every time I visit there!

Figure 6: Sentiment analysis example: The color scheme shows the spectrum from positiveness (green) to negativeness (red) output by the LSTM model. Our model is able to read the first two sentences and get the correct prediction (positive), while the full-reading LSTM model gets confused and outputs the wrong label (negative).

computational cost. Spatially adaptive computation time Figurnov et al. (2016) was proposed for image classification and object detection tasks. Compared to their work, our model is powerful by utilizing the combinatorial complexity of actions.

Attention mechanism applied to text data are related. ReasonNet Shen et al. (2017) trains a policy module to determine whether to stop before accessing the full text on question-answering tasks. Similarly, the model of Dulac-Arnold et al. (2011) performs early-stopping on text classification tasks. Comparing with these related work, our proposed model's skimming and rereading mechanisms are innovative. In addition, Choi et al. (2016) and Lei et al. (2016) propose to select the relevant sentences which are critical for question answering and sentiment analysis, respectively. Their methods utilize prediction accuracy as the reward signal to train the policy module. However, in our work, the policy module is trained considering both accuracy and computational cost explicitly.

Other ways to reduce the inference computational cost for new examples have been considered. Bengio et al. (2015) proposes a scheme to selectively activate parts of the network. Bolukbasi et al. (2017) presents two schemes to adaptively utilize the network during inference: Given each data point, they first select a network and then select some components of that network.

One closely related work is Yu et al. (2017). The authors train their policy network end-to-end with reinforcement learning. In contrast to their work, our model implemented human-like mechanism rereading and separated early-stopping mechanism, thus leading to further improved efficiency and accuracy. Furthermore, we hardly rely on many hyper-parameters and only use a simple reward structure. Finally, we get an advanced performance with better reward design which incorporates the negative energy cost explicitly and implement a value network to reduce the variance.

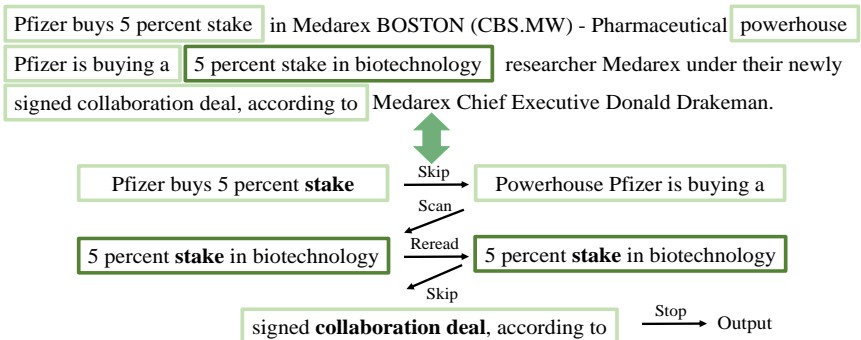

Figure 7: Topic classification example: Our model skips irrelevant chunks and reread an important phrase with "stake." It also early stops after reading "collaboration deal" and outputs the right classification (Business), but the full-reading model is fooled to output a wrong label (Technology).

## 5 CONCLUSIONS

We develop an end-to-end trainable approach for skimming, rereading and early stopping applicable to classification tasks. By mimicking human fast reading, we introduce a policy module to decide what token to read next (e.g., rereading the current token, reading the next one, or skipping the next few tokens) or whether the model should stop reading and form a decision. To encourage fast and accurate reading, we incorporate both classification accuracy and the computational cost as a reward function to score classification or other actions made by the agent during reading. An end-to-end training algorithm based on the policy gradient method backpropagates the reward signal into both the policy module (also including the classification policy) and the recurrent encoder. We demonstrate the efficacy of the proposed approach on four different datasets and demonstrate improvements for both accuracy and computational performance.

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

# 6 APPENDIX

## 6.A COMPARISON OF DIFFERENT CHUNK SIZE

To illustrate that our model's performance is robust to the choice of chunk size, we investigate the model performance with a variety of chunk sizes on the IMDB dataset. The result is shown in Figure 8. Here the red curve denotes the performance of the partial reading baseline, and the other three curves denote the performance of our full-action model with three chunk sizes 8, 20, 40, respectively. It is clear that our model outperformes the baselines significantly with different choices of chunk size.

In addition, we found that if the chunk size is too small, there are more decision steps inside each sentence, resulting the policy optimization more difficult. For instance, the performance of the chunk size 8 seems worse than two larger chunk sizes. We believe this issue may be overcome by applying more advanced policy optimization algorithms such as proximal policy optimization (Schulman et al., 2017). On the other hand, if the chunk size is too large, there were fewer decision steps, making the model not flexible enough. Among all three choices, we found that setting chunk size 20 leads to best practice in our experiments.

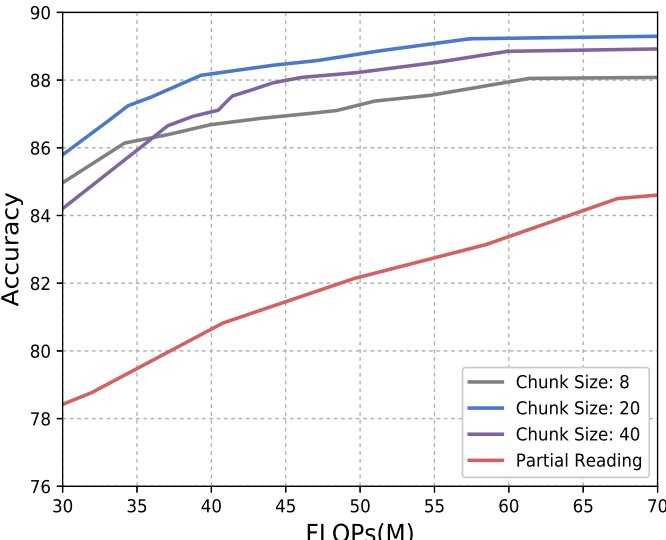

Figure 8: Comparison between different chunk sizes: Here the x-axis and y-axis are the same as previous figures. The red curve denotes the partial reading baseline, while the grey, blue, purple curves denote our models with chunk size 8, 20, 40, respectively. We found that our model is robust to different chunk sizes.

