# OpenReview forum: "Fast and Accurate Text Classification: Skimming, Rereading and Early Stopping"
_ICLR.cc/2018/Conference — Invite to Workshop Track_

### Official Review · AnonReviewer3 · 2017-11-27
**Interesting model but strong lacks in related works**

**Rating:** 5
**Confidence:** 3

**Review:**

The authors propose a sequential algorithm that tackles text classification while introducing the ability to stop the reading when the decision to make is confident enough. This sequential framework -reinforcement learning with budget constraint- will be applied to document classification tasks. The authors propose a unified framework enabling the recurrent network to reread or skip some parts of a document. Then, the authors describe the process to ensure both a good classification ability & a reduced budget.
Experiments are conducted on the IMDB dataset and the authors demonstrate the interest to select the relevant part of the document to make their decision. They improve both the accuracy & decision budget.


In the architecture, fig 1, it is strange to see that the decision to stop is taken before considering the label probability distribution. This choice is probably made to fit with classical sequential decision algorithms, assuming that the confidence level can be extracted from the latent representation... However, it should be discussed.

The interest of rereading a word/sentence is not clear for me: we simply choose to overweight the recent past wrt the further. Can it be seen as a way to overcome a weakness in the information extraction?

At the end of page 4, the difference between the early stopping model & partial reading model is not clear for me. How can the partial reading model be overcome by the early-stopping approach? They operate on the same data, with the same computational cost (ie with the same algorithm?)

At the end of page 6, authors claim that their advantage over Yu et al. 2017 comes from their rereading & early stopping abilities:
- given the length of the reviews may the K-skip ability of Yu et al. 2017 be seen as an early stopping approach?
- are the authors confident about the implementation of the Yu et al. 2017' strategy?
- Regarding the re-reading ability: the experimental section is very poor and we wonder:
  -- how the performance is impacted by rereading?
  -- how many time does the algorithm choose to reread?
  -- experiments on early-stopping VS early-stopping + skipping + rereading are interesting... We want to see the impact of the other aspects of the contribution.

On the sentiment analysis task, how does the model behave wrt the state of the art?

Given the chosen tasks, this work should be compared to the beermind system:
http://deepx.ucsd.edu/#/home/beermind
and the associated publication
http://arxiv.org/pdf/1511.03683.pdf
But the authors should also refer to previous work on their topic:
https://arxiv.org/pdf/1107.1322
The above mentioned reference is really close to their work.


This article describes an interesting approach but its main weakness resides in the lack of positioning wrt the literature and the lack of comparison with state-of-the-art models on the considered tasks.

---

> ### Author Response · Authors · 2017-12-30
> **Response to AnonReviewer3. Thank you for your comments!**
>
> 1- Concerning the model architecture: To save computation, our policy module estimates an action based upon the latent representation rather than the prediction. We assumed the label probability and confidence level are encoded in this representation. We addressed this in our revision.
>
> 2- Concerning the intuition for rereading: Our rereading mechanism was inspired by the success of bi-directional RNNs and human reading. For example, when reading an article, we may need to reread the previous paragraph to obtain a better understanding. Different from overweighting the previous data, we point out that weighting is dependent on the current context.
>
> 3- Regarding the comparison between early-stopping and partial reading: In our experiments, we firstly trained an early-stopping model and obtained the truncated sentence for both training and test set. Based on the truncated dataset, we trained a partial reading model. Thus, although these two models have the same computational cost, they are trained with different datasets. We discuss this setting in our revision.
>
> 4- Concerning the advantage over Yu et al. 2017: Firstly, the K-skip ability of Yu et al. can be viewed as a combination of skimming and early-stopping. However, their approach adopted classification accuracy as a reward function, while we utilized both accuracy and computation cost together, leading to a more comprehensive understanding of the accuracy-computation trade-off. Secondly, based on communication with Yu et al., we are confident in our implementation.
> Thirdly, we conducted an ablation study to show the effectiveness of each component (see Subsection 3.3 and Figure 7). We observe skimming, rereading, and the combination of both to improve the performance.
>
> 5- Regarding the comparison to other state-of-the-art methods: To the best of our knowledge, our RNN performance is comparable to recent work on four standard datasets with similar data processing and RNN architecture:
> IMDB: 89.1 ([1]), AG_news: 88.1 ([1]), Yelp: 94.74 ([2]), DBpedia: 86.36 ([3])
>
> [1]. Learn to skim Text
> [2]. Character-level convolutional network for text classification
> [3]. Semi-supervised sequence learning

---

### Official Review · AnonReviewer2 · 2017-11-27
**Unclear focus: is it the speed gains or the improved accuracy? Experimental comparison not convincing**

**Rating:** 5
**Confidence:** 4

**Review:**

This paper proposes to augment RNNs for text classification with a mechanism that decides whether the RNN should re-read a token, skip a number of tokens, or stop and output the prediction. The motivation is that one can stop reading before the end of the text and/or skip some words and still arrive to the same answer but faster.

The idea is intriguing, even though not entirely novel. Apart from the Yu et al. (2017) cited, there is older work trying to save computational time in NLP, e.g.:
Dynamic Feature Selection for Dependency Parsing.
He He, Hal Daumé III and Jason Eisner.
Empirical Methods in Natural Language Processing (EMNLP), 2013
that decides whether to extract a feature or not.
However, what is not clear to me what is achieved here. In the example shown in Figure 5 it seems like what happens is that by not reading the whole text the model avoids passages that might be confusing it. This could improve predictive accuracy (as it seems to do), as long as the model can handle better the earlier part of the text. But this is just an assumption, which is not guaranteed in any way. It could be that the earlier parts of the text are hard for the model. In a way, it feels more like we are addressing a limitation of RNN models in understanding text.

Pros:
- The idea is intersting and if evaluated thoroughly it could be quite influential.

Cons:
- the introduction states that there are two kinds of NLP problems, sequence2sequence and sequence2scalar. I found this rather confusing since text classification falls in the latter presumably, but the output is a label. Similarly, PoS tagging has a linear chain as its output, can't see why it is sequence2scalar. I think there is a confusion between the methods used for a task, and the task itself. Being able to apply a sequence-based model to a task, doesn't make it sequential necessarily.

- the comparison in terms of FLOPs is a good idea. But wouldn't the relative savings depend on the architecture used for the RNN and the RL agent? E.g. it could be that the RL network is more complicated and using it costs more than what it saves in the RNN operations.

- While table 2 reports the savings vs the full reading model, we don't know how much worse the model got for these savings.

- Having a trade-off between savings and accuracy is a good idea too. I would have liked to see an experiment showing how many FLOPs we can save for the same performance, which should be achievable by adjusting the alpha parameter.

- The experiments are conducted on previously published datasets. It would be good to have some previously published results on them to get a sense of how good the RNN model used is.

- Why not use smaller chunks? 20 words or one sentence at the time is rather coarse. If anything, it should help the model proposed achieve greater savings. How much does the choice of chunk matter?

- it is stated in the conclusion that the advantage actor-critic used is beneficial, however no experimental comparison is shown. Was it used for the Yu et al. baseline too?

- It is stated that model hardly relies on any hyperparameters; in comparison to what? It is better to quantify such statements,

---

> ### Author Response · Authors · 2017-12-30
> **Response to AnonReviewer2. Thank you for your comments!**
>
> 1- Regarding the FLOP counts: The FLOP counts, already provided in the first version, demonstrate that RL nets with significantly fewer operations are sufficient. To provide additional information, take the IMDB dataset as an example: the average number of read words is about 100, so the average number of decisions is 100 / 20 = 5 per sentence (chunk size is 20). The computational cost for each decision is 0.2 million FLOPs. The FLOP count for all decisions is hence 0.2 * 5 = 1 million FLOPs per sentence. Note that this cost is much smaller than the cost of the classifier (at least 25 million FLOPs as shown in Figure 2).
>
> 2- Comparison between our model and full-reading: In Table 2, we compared our proposed model to the one of [1] by presenting the energy cost necessary to achieve identical accuracy. Our model only needs 29.33% FLOPs, while 57.80% are needed for the model of [1]. By adjusting the computational budget, our optimal model can achieve a 0.5-1% accuracy improvement compared to the full-reading baseline.
>
> 3- FLOPs saved for the same performance: In addition to Table 2, please see our results in Table 1. We achieve a 4.11x, 1.85x, 2.42x, 1.58x speedup compared to the full-reading baseline on four datasets. Here, 4.11x means that our proposed model only needs 1/4.11 times the energy.
>
> 4- Regarding the baselines: To the best of our knowledge, our RNN performance is comparable to recent work on four standard datasets with similar data processing and RNN architecture:
> IMDB: 89.1 ([1]), AG_news: 88.1 ([1]), Yelp_polarity: 94.74 ([2]), DBpedia: 86.36 ([3])
>
> 5- Concerning the chunk size: To show the performance of different chunk sizes (8, 20, 40), we conducted experiments on the IMDB dataset. The experimental result has been added to the Appendix (Figure 8). We observe our proposed method to outperform the partial reading baseline with a significant margin. Notice that a smaller chunk size leads to a larger number of decision steps for each sentence, resulting in a complicated problem for policy optimization. Thus, prediction accuracy is slightly worse compared to a large chunk size. However, we believe this could be overcome by applying more advanced policy optimization algorithms like proximal policy optimization, left for future work. On the other hand, if the chunk size is too large (40), few decision steps inside the sentence hardly capture differences.
>
> 6- Advantage actor-critic use: Advantage actor-critic builds upon REINFORCE. For fair comparison, our training algorithm is the same as the algorithm used by Yu et al [1].
>
> 7- Hyper-parameters:  We only have a single hyper-parameter alpha to control the budget of the model. In contrast, Yu et al. use three hyper-parameters (N, K, R) to control the budget limitation.
>
> [1]. Learn to skim Text
> [2]. Character-level convolutional network for text classification
> [3]. Semi-supervised sequence learning

---

> > ### Comment · AnonReviewer2 · 2018-01-11
> > **Response to author response**
> >
> > The authors addressed most of my comments. However the didn't address the following:
> > - "the introduction states that there are two kinds of NLP problems, sequence2sequence and sequence2scalar...."
> > - "Actor-critic use": it is still stated the this paper illustrated the advantage of this approach, but it turns out that this is the same as Yu et al. thus it is not novel as implied in the last paragraph before conclusions.
> > Also, it seems like the choice of chunk size is an important hyperparameter of the model, since it affects its efficiency in terms of savings. This needs to be explored fully. Also Figure 8 is referenced as also doing component analysis in section 3.3, and each curve uses different components of the model. Is it meant to be Figure 7 there?
> > - There should be more comparisons with the very closely related method of Yu et al. (2017), but almost all of them are against the partial reading baseline. It is odd to consider Yu et al. only in Table 2 and only 2 datasets, and not  throughout the paper, in order to convince of the novelty and impact of the proposed approach.
> >
> > I have improved my score, but I still think the ready is not at the required level for publication in ICLR.

---

> > > ### Author Response · Authors · 2018-01-21
> > > **Response to AnonReviewer2. Thanks again for your comments!**
> > >
> > > Thanks for the insightful reviews, here are our answers to these questions:
> > > 1- Concerning the sequence2sequence and sequence2scalar problem: We have a typo here that writing the PoS tagging as "seq2scalar", this task should be classification problem. We are only addressing the classification problem in this work since it is complicated to directly apply skimming and early-stopping mechanisms into seq2seq task since the model should at least visit each unit in the seq2seq tasks. We will leave this research question as interesting further work.
> > > 2- Concerning the use of actor-critic: Thanks for pointing out this statement. We have addressed the issue in our revision. The new statement is that we argue the advanced performance is brought by a better reward design which incorporates the negative energy cost explicitly.
> > > 3- Concerning the chunk size: Yes, the reference should be figure 7 rather than figure 8 here. Also, we notice the chunk size here is as same as the `frame-skip` hyper-parameter in conventional reinforcement learning. So automatically choosing the best frame-skip remains an interesting future work, both for RL and its NLP application.
> > > 4- The reason we are using these two datasets is that Yu et al. only performs experiments on those two. We then conducted experiments on the whole four datasets and updated our result in the revision. On word-level DBpedia dataset, we selected the number of tokens before a jump as five as it is the same as the chunk size of our experiment. Then we conducted a grid search on hyper-parameters (N, K) and finally chose the one with better accuracy than full reading baseline. Here N=8 and K = 3. Yu et al.'s relative FLOPs is 76.36% while ours is 44.34% under same accuracy. Thus, our model outperforms Yu et al.'s with a large margin. Similarly, we obtained the optimal hyper-parameters N = 15 and K = 3 on sentence-level Yelp dataset. Here Yu et al.'s relative FLOPs is 82.34%, which is higher than our FLOPs 70.02% under same accuracy. So our model outperforms Yu's paper on all four datasets, making the experimental result more convincing.

---

### Official Review · AnonReviewer1 · 2017-11-28
**You don't need to read the entire review to classify it.**

**Rating:** 7
**Confidence:** 4

**Review:**

The paper present a model for fast reading for text classification with mechanisms that allow the model to reread, skip words, or classify early before reading the entire review. The model contains a policy module that makes decisions on whether to reread, skim or stop, which is rewarded for both classification accuracy and computation cost. The entire architecture is trained end-to-end with backpropagation, Monte Carlo rollouts and a baseline for variance reduction.

The results show that the architecture is able to classify accurately on all syntactic levels, faster than a baseline that reads the entire text. The approach is simple and seems to work well and could be applied to other tasks where inference time is important.

---

> ### Author Response · Authors · 2017-12-30
> **Response to AnonReviewer1. Thank you for your comments!**
>
> We added more details to illustrate the effectiveness of our proposed model. We envision this policy mechanism to be helpful for more general tasks beyond language understanding. We leave exploration of those to future work.

---

### Public Comment · ~Rahul_Ravu1 · 2017-10-30
**Related Work**

The paper referenced in Related Work, "Rationalizing Neural Predictions" seems a bit similar to the current work. Also, the comment that they use attention for generating rationales is a bit confusing as the encoder which generates the label only sees the rationales(subset of the original text unless I am mistaken).

---

### Public Comment · (anonymous) · 2017-11-27
**Early stopping**

The early stopping idea is already implemented in the related work "Learning to Skim Text", i.e., the reading will stop if 0 is sampled from the jumping softmax. This can be seen in the two examples of their last experiment.

---

### Public Comment · (anonymous) · 2017-11-27
**Related Workd**

''Text Classification: A Sequential Reading Approach.'' published in 2011 is also clearly related.

---

### Author Response · Authors · 2017-12-30
**General response to the reviewers**

We thank the reviewers for their feedback and address their comments below. All modifications are highlighted (blue) in the newly uploaded version. The main modifications:
1. Added more details for our experimental setup
2. Added an ablation study to demonstrate the effectiveness of each component in our proposed model
3. Added more details for the comparison between our proposed model and Yu et al. 2017

---

### Decision · Program_Chairs · 2018-01-29
**ICLR 2018 Conference Acceptance Decision**

**Decision:**

Invite to Workshop Track

**Comment:**

this is an interesting approach that applies the idea of dynamically controlling the amount of information from the input fed into the classifier (some of the earlier approaches have used this idea for, e.g., parsing, real-time translation, online speech recognition, and so on...) this is also related to some of the recent work on hierarchical recurrent nets [Chung et al.]. unfortunately, two of the reviewers and other commenters found this manuscript needs more work to clarify motivation, implication and relationship to other existing works, with which i don't necessarily disagree.